# Application of CRISPR/Cas9 for Rapid Genome Editing of Pseudorabies Virus and Bovine Herpesvirus-1

**DOI:** 10.3390/v16020311

**Published:** 2024-02-18

**Authors:** Wanqi Yu, Jingyi Liu, Yingnan Liu, Maria Forlenza, Hongjun Chen

**Affiliations:** 1Shanghai Veterinary Research Institute, Chinese Academy of Agricultural Sciences, Shanghai 200241, China; wanqi.yu@wur.nl (W.Y.); liujingyi@shvri.ac.cn (J.L.); liuyingnan@shvri.ac.cn (Y.L.); 2Institute of Animal Sciences, Wageningen University and Research, 6708 WD Wageningen, The Netherlands

**Keywords:** herpesvirus, recombination, gene editing, CRISPR/Cas9

## Abstract

The CRISPR/Cas9 system is widely used to manipulate viral genomes. Although *Alphaherpesvirinae* genomes are large and complicated to edit, in recent years several Pseudorabies virus (PRV) mutants have been successfully generated using the CRISPR/Cas9 system. However, the application of CRISPR/Cas9 editing on another member of alpha herpesviruses, bovine herpesvirus-1 (BHV-1), is rarely reported. This paper reports a rapid and straightforward approach to manipulating herpesviruses genome using CRISPR/Cas9. The recombinant plasmids contained the left and right arm of the *thymidine kinase* (*TK*) gene of PRV or of the *glycoprotein I* (*gI*) and *glycoprotein E* (*gE*) of BHV-1. Upon the cleavage of the TK or gIgE gene by Cas9 protein, this was replaced by the *enhanced green fluorescence protein* (*eGFP*) by homologous recombination. With this approach, we generated recombinant TK-/eGFP+ PRV and gIgE-/eGFP+ BHV-1 mutants and then proceeded to characterize their biological activities in vitro and in vivo. In conclusion, we showed that alpha herpesvirus, including PRV and BHV-1, can be rapidly edited using the CRISPR/Cas9 approach paving the way to the development of animal herpesvirus vaccines.

## 1. Introduction

The *alphaherpesvirinae* subfamily contains human herpes simplex viruses (HSV) types 1 and 2, simian agent 8 (SA8) virus, monkey B virus [1], bovine herpesviruses (BHV), pseudorabies virus (PRV), equine herpesvirus (EHV), varicella-zoster virus (VZV) and Marek’s disease viruses (MDV). The members of this subfamily can infect various animal species, have a short replication cycle, spread quickly and can establish lifelong latent infections [2]. Among them, animal herpes viruses cause substantial economic losses to husbandry [3]. In recent years, new virulent strains have emerged [4,5,6], and it is crucial to design novel vaccines to control them. In addition, owing to the genetic stability of their large double-stranded DNA (dsDNA) genome, and the presence of many non-essential genes that can be replaced by gene insertions [7], *alphaherpesvirinae* can serve as the backbone of multivalent vectored vaccines.

Gene editing is widely used to obtain attenuated and/or recombinant vaccine candidates through a variety of technologies: homologous recombination [8], zinc-finger nucleases (ZFNs) [9], transcription activator-like effector nucleases (TALENs) [10] and clustered regularly interspaced short palindromic repeats (CRISPR)/CRISPR-associated protein 9 (Cas9) [11]. Due to its ease of use, CRISPR/Cas9 has become the first choice for gene editing. CRISPR is a family of repetitive DNA sequences found in the genomes of prokaryotes such as bacteria and archaea [12]. These sequences are the remnants of previous viral infections and can be used by the bacterium to detect DNA from similar viruses during subsequent infections. A CRISPR sequence can guide the Cas9 enzyme to recognize and cut specific DNA strands complementary to the CRISPR sequence. The CRISPR/Cas9 editing process has a wide range of applications, including fundamental research, the development of biotechnological products, and the treatment of diseases [13]. CRISPR/Cas9 can also be used to knock out virulence genes of a virus of interest and cooperate with the homologous recombination machinery to achieve site-directed insertion of a foreign gene.

Currently, mammalian herpesvirus-based attenuated vaccines most commonly use the backbone of two viruses: PRV and bovine herpesvirus-1 (BHV-1). PRV, also known as Aujeszky’s disease virus (ADV) or porcine herpesvirus-1 (SuHV-1), can infect not only its natural host, pigs, but also cattle, sheep, rabbits and mice [14]. A small number of human infections with PRV have also been reported in recent years [15]. In pigs, the PRV Bartha K61 strain can trigger a wide range of humoral and cellular immune responses as well as be a safe and effective multivalent vaccine backbone [16]. Bartha K61 recombining the *GP5* gene of Porcine Reproductive and Respiratory Syndrome Virus (PRRSV) has been proved to reduce pathogenic lesions caused by PRRSV infection in vaccinated pigs [17]. BHV-1, also known as infectious bovine rhinotracheitis virus (IBRV), can cause infectious pustular bursitis (IPB) in bulls and infectious pustular vulvovaginitis (IPV) in cows. Recombinant BHV-1 expressing the *E2* protein of bovine viral diarrhea virus (BVDV) could effectively prevent the infection of two viruses [18]. And the study of bighorn sheep against Mannheimia haemolytica with a BHV-1-vectored vaccine indicated that BHV-1 is an essential vector for the immunization of wild animals [19]. Although herpesvirus genomes are complicated to edit, several PRV mutants have been successfully generated using the CRISPR/Cas9 system in recent years [20,21,22]. However, the application of CRISPR/Cas9 editing to BHV-1 has not been extensively investigated. Due to the low transfection efficiency of cell lines that allow BHV-1 to replicate, it is cumbersome [23,24] and slow to construct a recombinant BHV-1 mutant.

In this study, we used specific approaches to edit different herpes virus (PRV and BHV-1) genomes via CRISPR/Cas9 and homologous recombination. The *enhanced green fluorescence protein* (*eGFP*) gene was used to replace the *thymidine kinase* (*TK*) gene of PRV or the *glycoprotein I* (*gI*) and *glycoprotein E* (*gE*) genes of BHV-1. The *TK* gene, also called *unique long region (UL) 23*, was targeted because *TK*-deficient mutants proved to be highly attenuated in mice and pigs [25,26]. And the *gI* and *gE* genes, also called unique short region (US) 7 and US8, were targeted since they are complexed with each other and contribute to virulence [27,28]. The biological characteristics of the obtained candidate vaccine strains were also evaluated in vitro and in vivo. We have established a technology platform for alphaherpesvirus gene editing that can be used to rapidly construct genetically engineered anti-viral vaccines.

## 2. Materials and Methods

### 2.1. Cell Lines and Viruses

Human embryonic kidney (HEK293T) cells, baby hamster kidney fibroblast (BHK-21) cells and Madin–Darby bovine kidney (MDBK) cells were purchased from American Type Culture Collection (ATCC, Gaithersburg, MD, USA ). All cell lines were maintained in Dulbecco’s modified medium (DMEM, Gibco, Grand Island, NY, USA) supplemented with 10% fetal bovine serum (FBS, Gibco, Grand Island, NY, USA), 0.1 mg/mL streptomycin and 100 IU/mL penicillin (Gibco, Grand Island, NY, USA). The PRV Bartha K61 strain was purchased from Weike Biotech Co., Harbin, China. The BHV-1 Bartha Nu67 strain was purchased from China Veterinary Culture Collection Center (CVCC, Beijing, China).

### 2.2. Construction of Recombinant Plasmid and Guide RNAs

The recombinant plasmid pUC-TKLR-eGFP was constructed as illustrated in Figure 1A. The left and right arms (termed TK-L arm and TK-R arm) were amplified using polymerase chain reaction (PCR) using Phanta Max Super-Fidelity DNA Polymerase (Vazyme, Nanjing, China) and 2× GC Buffer (Takara, Beijing, China) from PRV (GenBank accession number: JF797217.1) genomic DNA. It should be noted that the TK-L arm contains a previously reported *TK* promotor (TKp) [29]. The eGFP was amplified using PCR from the pEGFP-C1 plasmid. The TK-L arm, eGFP and TK-R arm were then ligated by using overlapping PCR and cloned into pUC19 plasmid through *Eco*R I (NEB, Ipswich, MA, USA) and *Hin*d III (NEB, Ipswich, MA, USA) restriction sites.

The recombinant plasmid pUC-gIELR-eGFP was constructed as illustrated in Figure 1B. The left and right arms (termed gIE-L arm and gIE-R arm) were amplified using PCR from BHV-1 (GenBank accession number: KU198480.2) genomic DNA. The CMV promoter (CMVp)-eGFP box was amplified from the pEGFP-C1 plasmid. The gIE-L arm, CMVp-eGFP and gIE-R arm were then ligated using overlapping PCR. This PCR product was cloned into the *Ec*oR I and *Hind III* -linearized pUC19 plasmid by using the ClonExpressII One Step enzyme (Vazyme, Nanjing, China).

The guide RNAs (sgRNAs) targeting the *TK* gene, and *gI* or *gE* genes were designed using the CRISPR Design Tool (https://zlab.bio/guide-design-resources, accessed on 25 October 2017). Two guide RNA sequences were chosen upstream and downstream of the target region (Figure 1A,B). All guide RNAs were separately synthesized and cloned into a *Bbs I* (NEB, Ipswich, MA, USA)-digested pX335 plasmid (Addgene plasmid catalog: 42335). All used primers are listed in Table 1.

### 2.3. Cell Transfection

HEK293T cells were seeded in a 6-well plate with 5 × 10^5^ cells per well and the next day transfected with 2 μg recombinant pUC19-TKLR-eGFP plasmid using TransIT-LT1 Transfection Reagent (Mirusbio, Madison, WI, USA) following the manufacturer’s instruction (with a plasmid: reagent ratio of 1:2.5). The activity of the TK promoter was evaluated by *eGFP* expression 24 h after transfection.

BHK-21 cells were seeded in a 6-well plate with 5 × 10^5^ cells per well and transfected with 1 μg PRV genomic DNA using TransIT-LT1 Transfection Reagent (with a plasmid: reagent ratio of 1:2.5). Cytopathic effects (CPE) were recorded 48 h after transfection.

### 2.4. Generation of Virus Mutants

BHK-21 cells were seeded in a 6-well plate with 5 × 10^5^ cells per well and co-transfected with 2 μg PRV Bartha K61 genomic DNA, 1 μg pUC19-TKLR-eGFP plasmid, 1 μg pX335-TK1 plasmid and 1 μg pX335-sgRNA-TK2 plasmid. The cells were monitored every 12 h post-transfection. When CPE occurred, the cell culture was collected and subjected to three freeze–thaw cycles. The supernatant containing recombinant PRV mutant was harvested after centrifugation at 13,400× *g* for 5 min.

HEK293T cells were seeded in a 6-well plate with 5 × 10^5^ cells per well and co-transfected with 1 μg pUC-gIELR-eGFP plasmid, 1 μg pX335-gIE1 plasmid and 1 μg pX335-gIE2 plasmid. Simultaneously, the cells were infected with 20 μL of BHV-1 Bartha Nu67 virus (TCID50 = 5 × 10^7^). After 48 h, the cells were transferred to a 100 mm dish already containing a confluent monolayer of MDBK cells. When CPE occurred in MDBK cells, the supernatant containing recombinant BHV-1 mutant was collected as described above.

### 2.5. Plaque Purification

The procedure for plaque isolation is summarized in Figure 2. BHK-21 and MDBK cells were grown to confluency in 100 mm dishes and infected with a 10 times serial dilution of PRV or BHV-1 mutant for 2 h at 37 °C, respectively. The cells were overlaid with a mixture of 2% low melting point agarose and 2 × DMEM (Procell, Wuhan, China) containing 2% FBS at 1:1 ratio and incubated at 37 °C under 5% CO_2_. The plates were observed every 12 h, until green fluorescence was visible. Single plaques of *eGFP* positive (*eGFP*+) cells were then transferred to the 96-well plate on BHK-21 or MDBK monolayers. When *eGFP* expression was detected, cells and supernatant were collected for the next round of plaque purification. This procedure was repeated twice. The obtained recombinant viruses were termed PRV TK-/eGFP+(PRVmu) and BHV-1 gIE-/eGFP+(BHVmu).

### 2.6. Identification of Recombinant Viruses by PCR

The PRVmu and BHVmu genomic DNA was extracted using a TIANamp Genomic DNA kit (TIANGEN, Beijing, China) according to the manufacturer’s instructions, and used for the PCR amplification of the inserted fragments using primers listed in Table 1 (TK-check, eGFP-check, gI-check and gE-check primers). The PCR products were sequenced using Sanger sequencing (GENEWIZ, Suzhou, China).

### 2.7. Growth Kinetics of Recombinant Viruses

BHK-21 and MDBK cells were used to determine the growth kinetics of the recombinant PRVmu and BHVmu viruses, respectively. The BHK-21 cells were infected with PRV Bartha K61 or PRVmu virus at 0.01 multiplicity of infection (moi). The MDBK cells were infected with BHV-1 Bartha Nu67 or BHVmu virus at 0.01 moi. At 8, 16, 24, 32, 40, 48, 56 and 64 h post-inoculation, the infected cell cultures were frozen and thawed three times. Then, the supernatant was collected for virus titration. The viral titers were measured as the median tissue culture infective dose (TCID_50_) according to the Reed–Muench method.

### 2.8. Virus Infectivity in Mice

Five-week-old BALB/c mice (purchased from SPF-Biotech, Beijing, China) were randomly divided into six groups (Groups I-VI). Groups I-III (n = 9 per group) were inoculated intranasally with 50 µL 5 × 10^4^ TCID_50_/_mL_ PRV Bartha K61, 50 µL 5 × 10^4^ TCID_50_/_mL_ PRVmu or 50 µL DMEM as the negative control, respectively. Groups IV-VI (n = 8 per group) were inoculated with BHV-1 Bartha Nu67, BHVmu and DMEM using a similar approach as for Groups I-III. Mice in groups IV-VI were boosted with the same dose and route 7 days after the first immunization. The weight of the mice was recorded daily. At 14 days post-inoculation, all surviving mice were euthanized and the serum samples were collected for further analysis.

### 2.9. Detection of Virus-Specific Antibodies

The production of anti-BHV-1 *glycoprotein B* (*gB*) and anti-*gE* antibodies was evaluated using enzyme-linked immunosorbent assay (ELISA) (INgezim IBR Compac or gE Compac, Ingenasa, Madrid, Spain) according to the manufacturer’s instructions. Briefly, 50 µL/well diluent and 50 µL/well serum samples were added to the ELISA plates and incubated for 1 h at 37 °C. After several washes, the plates were incubated with peroxidase-conjugated anti-*gB* or anti-*gE* monoclonal antibodies (100 µL per well) for 30 min at 37 °C. After several washes, 100 µL/well substrate solution was added and incubated for 15 min at room temperature in the dark. Then, 100 µL/well stop solution was added and the absorbance was determined within 5 min using a microplate reader (BioTek, Irving, TX, USA) at 450 nm.

### 2.10. Statistical Analysis

Data were analyzed with GraphPad prism 7.0 (GraphPad software, USA). The ELISA results were valid when OD450 of the negative control >0.750 and OD450 of the positive control/OD450 of the negative control <0.250. The blocking percentage of anti-*gB* antibodies was calculated as (OD450 of the negative control—OD450 of the sample)/(OD450 of the negative control—OD450 of the positive control) × 100%. The blocking percentage of anti-*gE* antibodies was calculated as (1—(OD450 of the sample/OD450 of the negative control)) ×100%. The sample was determined as negative when the blocking rate was <25% and positive when the blocking rate was >30%. The body weight results were analyzed through two-way ANOVA followed by a Tukey post hoc test; *p* < 0.05 was considered statistically significant.

## 3. Results

### 3.1. Identification and Validation of TK Promotor and PRV Rescue from Genomic DNA

To investigate whether the TK promotor (TKp) that was cloned immediately upstream of eGFP was functional, we monitored eGFP expression (Figure 3A). The green fluorescence suggested that the inserted TKp was functional and could drive protein expression. Next, to investigate whether the PRV genome itself could be used to rescue the recombinant virus, we transfected PRV Bartha K61 genomic DNA into BHK21 cells. CPE were observed (Figure 3B) suggesting that infectious viruses can be rescued upon transfection of PRV genomic DNA alone. A similar approach using purified BHV-1 genome did not lead to virus rescue.

### 3.2. Construction of Recombinant Viruses Using CRISPR/Cas9 System

To obtain recombinant PRV, we co-transfected the pUC19-TKLR-eGFP plasmid, PRV Bartha K61 genomic DNA and CRISPR/Cas9 system components containing two sgRNAs, targeting the genomic regions flanking the *TK* gene into BHK-21 cells (Figure 4A). To obtain recombinant BHV-1, we co-transfected the recombinant pUC-gIELR-eGFP plasmid and two sgRNAs targeting the genomic regions flanking the *gIgE* gene into HEK293T cells, followed by infection with BHV-1 Bartha Nu67 virus. HEK293T cells treated in this way were then transferred onto a MDBK monolayer for recombinant virus replication. When CPE occurred (Figure 4B), BHK-21 cells or MDBK cells were collected for PCR verification with specific primers. The *eGFP* was inserted between 58,684 bp and 59,677 bp of PRV Bartha K61 genome leading to the complete removal of *TK* gene. The CMV promoter-eGFP box was inserted between 119,115 bp and 122,075 bp of BHV-1 Bartha Nu67 genome, which lead to the removal of the entire *gI* gene and 83.2% of the *gE* gene. The remaining *gE* sequence does not lead to the expression of a functional *gE* protein. After three rounds of plaque purification (Figure 4C), confirmation via PCR (Figure 4D) and Sanger sequencing, the purified recombinant PRV TK-/eGFP+ virus and the BHV-1 gIE−/eGFP+ virus were successfully constructed.

### 3.3. Growth Kinetics of the Recombinant Viruses

To determine whether *TK* deletion and *gIgE* deletion affect PRV and BHV-1 replication, respectively, we compared the replication of the recombinant and wild-type viruses. We found that the recombinant PRVmu virus replicated slower than the parental PRV in the first 24 h but ultimately reached the same titers (Figure 5A). The recombinant BHVmu virus displayed similar growth kinetics to the parental BHV-1 (Figure 5B).

### 3.4. Evaluation of Virus Attenuation

To evaluate the infectivity and virulence of PRVmu and BHVmu viruses in vivo, mice were infected with the recombinant viruses. To test whether the PRV TK-/eGFP+ was attenuated, we monitored the mice’s weight and survival. The results showed that the wild-type PRV Bartha K61 is lethal to mice (Figure 6A), causing notable weight loss in surviving mice at 6 and 7 days post infection (Figure 6B). However, the PRVmu-infected mice had a 78.75% survival rate (2 out of 9), with weight loss observed only in a few mice from 7 days post infection onward. These suggested that PRV was effectively attenuated by the deletion of the *TK* gene.

To test whether the BHV-1 gIE-/eGFP+ was attenuated, we measured the weight and the serum antibodies production of infected mice. In this experiment, all infected mice survived. Small differences in mouse body weight were observed only at 2 days post infection between the wild-type and the non-infected or BHVmu-infected group (Figure 6C).

Serological analysis showed that anti-*gE* antibodies were detected only in the serum of mice infected with BHV-1 Bartha Nu67 (Table 2). The absence of anti-*gE* antibodies in BHVmu-infected mice further confirmed that the *gE* gene successfully mutated leading to a lack of antigen expression. It should be noted that anti-*gB* antibodies were detected in the serum of mice infected with the wild-type BHV-1 and with the BHVmu virus. Since *gB* is the most highly conserved glycoprotein and an important protective antigenic protein in BHV-1 [30], this finding indicated that the recombinant BHVmu virus was still immunogenic and could trigger an immune response similar to that triggered by the wild-type virus.

These results revealed that the PRV and BHV-1 genome was successfully edited using CRISPR/Cas9 system, leading to recombinant mutants that can serve as safe potential vaccine candidates.

## 4. Discussion

Genetic manipulation is widely used to study virus biology and develop new vaccines. However, alphaherpesviruses are difficult to manipulate due to their large and complex genome. This study shows that the alphaherpesvirus genome can be rapidly edited using a CRISPR/Cas9 system, through the co-transfection of specific sgRNAs and recombinant plasmids.

Current methods for the construction of attenuated virus strains are based on the knock-out of one or more genes in the viral genome by homologous recombination. To increase the efficiency of traditional homologous recombination, we made use of the CRISPR/Cas9 system. CRISPR/Cas9 technology has been widely used in biological systems, such as yeast [31], insects [32], and mammals [33].

Despite its size, the PRV genome has been previously edited using CRISPR/Cas9 technology to knock out genes, generate live attenuated vaccines, and develop useful tools for virus biology studies [21,22,34,35]. Here, we have successfully deleted the *TK* gene of PRV by combining the CRISPR/Cas9 system with homologous recombination. Notably, we used the viral *TK* promoter to drive the expression of downstream reporter genes. Using the viral promoter instead of a foreign promoter not only shortens the length of the recombinant plasmid, making the plasmid easier to construct, but it also makes the generated PRVmu more suitable for subsequent applications as a recombinant vaccine vector. Despite the clear attenuation of the PRV mutant virus, it was observed that a small number of PRVmu-infected mice died. This could be due to the possibility that the PRVmu virus can use the host thymidine kinase, suggesting that the knockout of only the viral *TK* gene may not be sufficient to generate safe live recombinant PRV-based vaccines. Nevertheless, our data show that the knockout of viral *TK* can effectively reduce the pathogenicity of PRV, allowing most mice to survive and pave the way for the generation of subsequent live recombinant vectors.

Although the CRISPR/Cas9 system has been widely used to edit the genome of a wide range viruses, to our knowledge the present study is the first to report the generation of recombinant BHVmu virus combining the use of MDBK cells that are highly susceptible to BHV-1 with HEK293T cells that have a high transfection efficiency. By first transfecting recombinant plasmids and sgRNAs into HEK293T cells followed by BHV-1 infection and, when the CPE occurred, by transferring the HEK293T cells directly onto MDBK cells without freeze–thaw cycles, we greatly improved the efficiency of homologous recombination and expansion of recombinant virus in MDBK cells. This approach was described in a recent study [36] in which Liu et al. screened several cell lines for plasmid transfection and BHV-1 infection and found that HEK293T had the highest transfection efficiency (88.7%), followed by BHK-21 (66%) and Vero E6 cells (44.7%). However, HEK293T cells are not suitable for BHV-1 infection. Conversely, MDBK, Vero E6 and primary bovine testis cells (BT) can be used for subculture. These findings led to the use of HEK293T to generate a BHV-1 recombinant vector and the production of recombinant BHV-1 mutant, but the expansion of the latter was obtained using MDBK cells. When used in vivo, we found that the knockout of the *gI* and *gE* genes in our BHVmu virus retained the immunogenicity of BHV-1.

In conclusion, we successfully constructed a PRV-TK-/eGFP+ mutant and a BHV-gIgE-/eGFP+ mutant. The *eGFP* gene can be easily replaced with another target gene of interest using the same method, and the novel recombinant mutants can be used as multivalent vaccine candidates.

## Figures and Tables

**Figure 1 viruses-16-00311-f001:**
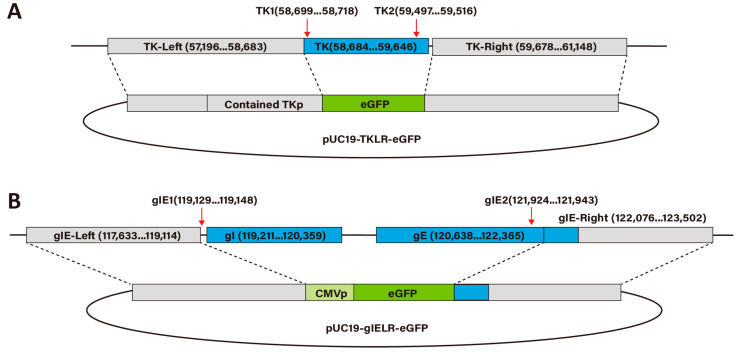
Construction of recombinant plasmids for the generation of PRV and BHV-1 recombinant mutants. (**A**,**B**) Schematic illustration of the donor plasmid sequence used to edit PRV and BHV-1 genome. Red arrows represent the sgRNA cleavage site on the viral genome; the numbers indicate the genomic regions flanking the target genes; regions between dashed lines indicate the left and right homology arms that were used to replace the gene of interest with the *eGFP* reporter. (**A**) For PRV, the left arm of the plasmid, upstream of the *eGFP* gene, contains the predicted endogenous *TK* promoter. (**B**) for BHV-1, since the endogenous promotor for the *gI* and *gE* gene is unknown, a constitutive CMV promotor was inserted directly upstream of the *eGFP* gene.

**Figure 2 viruses-16-00311-f002:**
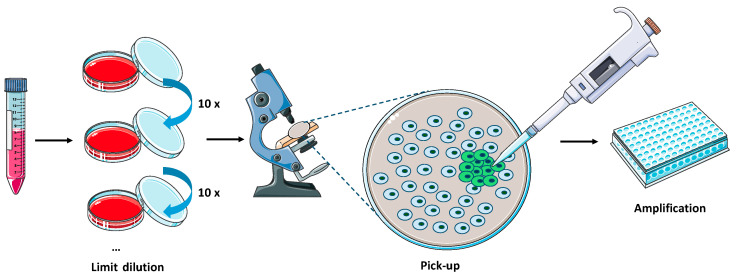
PRV and BHV-1 recombinant mutant rescue and flowchart of plaque purification. The supernatants containing wild-type and mutant viruses were serially diluted and used to infect BHK-21 or MDBK monolayers in 100 mm dishes. *eGFP*+ plaques were picked up upon visual identification under the microscope and transferred to 96-well plates with BHK-21 or MDBK monolayers for amplification.

**Figure 3 viruses-16-00311-f003:**
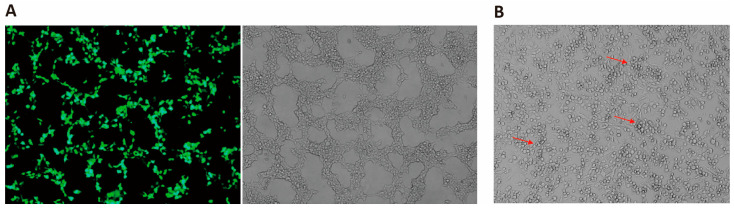
Identification and validation of *TK* promoter and PRV rescue from genomic DNA. (**A**) Recombinant pUC19-TKLR-EGFP plasmid (Figure 1A) was transfected into HEK293T cells and *eGFP* expression driven by the *TK* promotor was detected after 24 h. (**B**) PRV Bartha K61 genomic DNA was transfected into BHK21 cells. The red arrows indicate PRV plaques.

**Figure 4 viruses-16-00311-f004:**
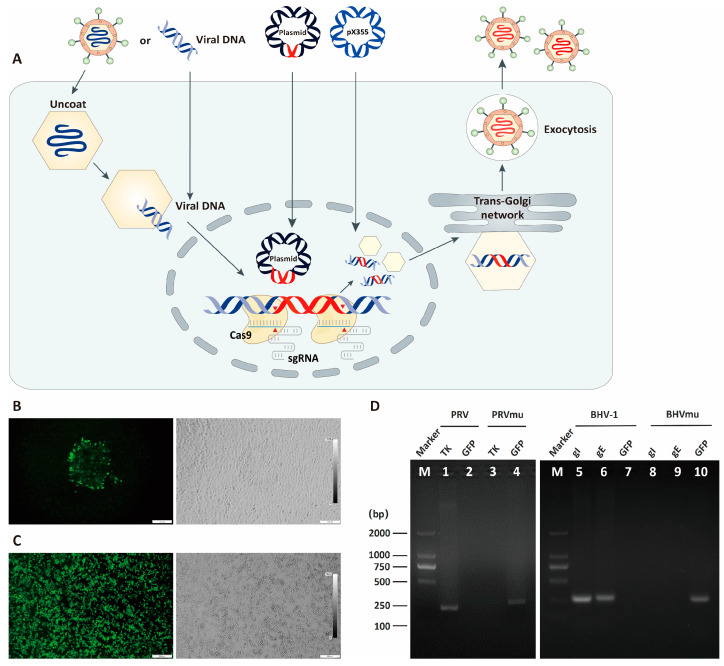
Construction and purification of recombinant PRV or BHV-1. (**A**) Schematic illustration of the strategy used to rescue PRV and BHV-1 mutants. The viral genomic DNA was transferred into cells via transfection or viral infection; the presence of CRISPR/Cas9 components, including the two sgRNA flanking the target gene, leads to genome editing in which the reporter gene present in the plasmid replaces the target gene in the viral genome by homologous recombination. (**B**) A plaque of BHV-1 mutant on MDBK cells. The HEK293T cells which contained wild-type BHV-1 and mutant BHV-1 were transferred onto MDBK monolayer, then CPE occurred. (**C**) Green fluorescence on infected MDBK cells. All cells were infected by the purified virus mutant. (**D**) PCR detection of PRV and BHV-1 mutant. The target genes were amplified in the wild-type viruses but not in the mutants.

**Figure 5 viruses-16-00311-f005:**
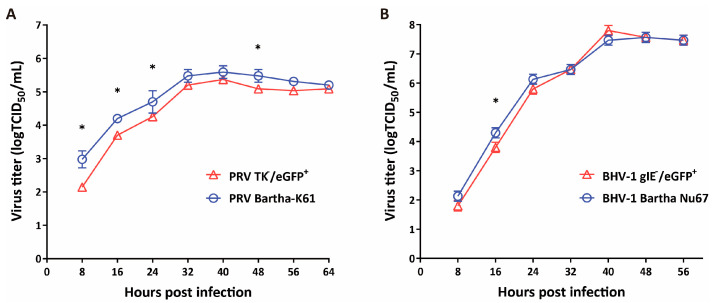
One step virus growth curves. The culture supernatants of (**A**) PRV and (**B**) BHV-1 were collected at the indicated time points and used to determine the viral titers. Data are presented as mean ± SD of three replicate samples measured in three independent experiments. * indicates significant differences (*p* < 0.05) as assessed through two-way ANOVA followed by Tukey post hoc test.

**Figure 6 viruses-16-00311-f006:**
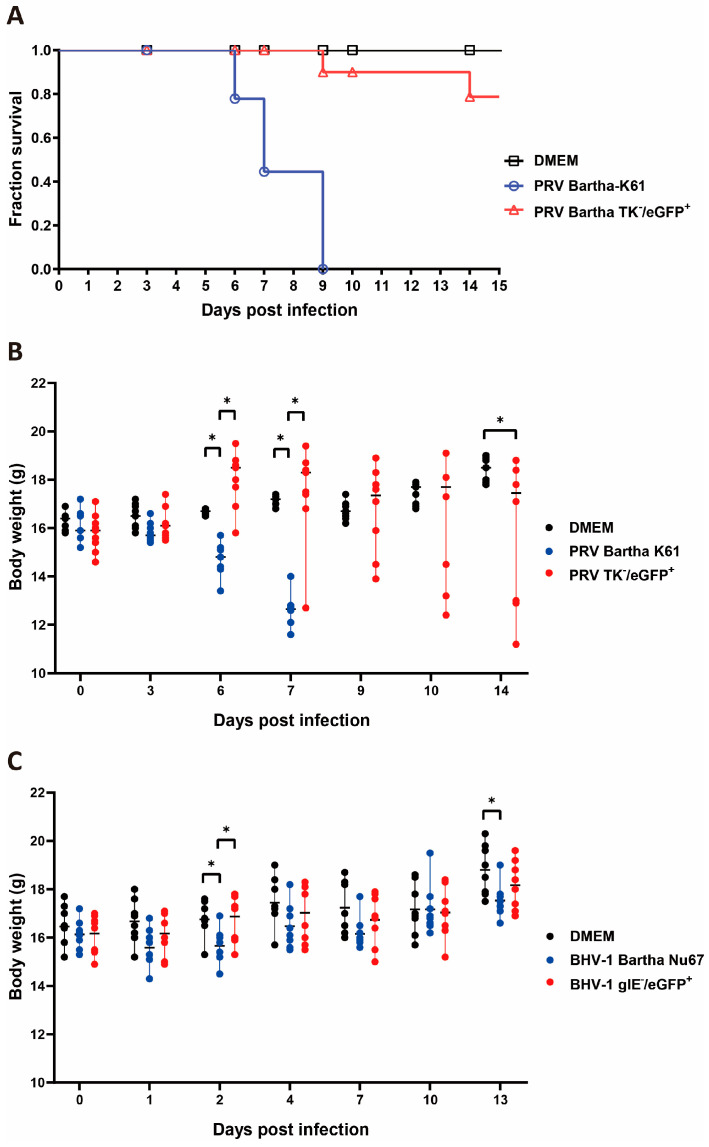
Mice weight and survival rate. (**A**) The survival fraction of mice after PRV Bartha K61 or PRVmu infection. The mice that lost more than 25% of their weight were considered dead. (**B**) The body weight of mice after PRV Bartha K61 or PRVmu infection. Data were presented as median from min to max with all the points. (**C**) The body weight of mice after BHV-1 Bartha Nu67 or BHVmu infection. Data were presented as median from min to max with all the points. The significance was analyzed via two-way ANOVA and shown with *.

**Table 1 viruses-16-00311-t001:** Primers and sgRNAs used in this study.

Primers and sgRNAs	Sequences (5′ to 3′)
TK-L-F	AAAACGACGGCCAGTGAATTCAGCACGCTGTGGCCCTCCAG
TK-L-R	CGCCCTTGCTCACCATATCCGCTGCCACAACCGCTTCTAC
TK-R-F	GACGAGCTGTACAAGTAAATGGAGACCGCGACGGAGGCAAC
TK-R-R	GACCATGATTACGCCAAGCTTAGGTTGGCCAGGGTGGCGTC
eGFP-F	CGCCCTTGCTCACCATCCCGGCGCGCTTCCGGGCGG
eGFP-R	GTTGCCTCCGTCGCGGTCTCCATTTACTTGTACAGCTCGTC
sgRNA-TK1-F	CACCGATCTACCTCGACGGCGCCTA
sgRNA-TK1-R	AAACTAGGCGCCGTCGAGGTAGATC
sgRNA-TK2-F	CACCGCGCGTCTCCACCGTCGACCT
sgRNA-TK2-R	AAACAGGTCGACGGTGGAGACGCGC
TK-check-F	TGGCCGGTATTTACGATGCG
TK-check-R	GCGCTGATGTCCCCGACGATG
eGFP-check-F	CAGTGCTTCAGCCGCTACCC
eGFP-check-R	TTCACCTTGATGCCGTTCTTC
gIE-L-F	AAAACGACGGCCAGTGAATTCGCGTTTACAATAAACAG
gIE-L-R	GATTACTATTAATAACTAGCTAGGAGCAAAGGGG
CMVp-eGFP-F	CCCCTTTGCTCCTAGCTAGTTATTAATAGTAATC
CMVp-eGFP-R	GATTACTATTAATAACTAGCTAGGAGCAAAGGGG
gIE-R-F	CCCCTTTGCTCCTAGCTAGTTATTAATAGTAATC
gIE-R-R	GACCATGATTACGCCAAGCTTACGGCGACGACGACGTGTTC
sgRNA-gIE1-F	CACCGATCTCCCGCCCCGCGCGGCT
sgRNA-gIE1-R	AAACAGCCGCGCGGGGCGGGAGATC
sgRNA-gIE2-F	CACCGCGCGCTTGGACTCGCGGGAC
sgRNA-gIE2-R	AAACGTCCCGCGAGTCCAAGCGCGC
gI-check-F	GTCGAGCTGCTGCGCTACCAC
gI-check-R	AAACGCGGCCAAGGGAAAGAC
gE-check-F	ACCTGCGTCCCGCCAATAAC
gE-check-R	ACCAGTCCCGCGAGTCCAAG

**Table 2 viruses-16-00311-t002:** Block ELISA OD values of gB and gE antibodies.

OD450			gB			gE		
Positive	0.1032 +	0.1028 +			0.0821 +	0.1091 +		
Negative	1.4816 −	1.4900 −			1.0886 −	1.0786 −		
DMEM	1.3086 −	1.3156 −			1.0434 −	0.9498 −		
BHV-1	0.2086 +	0.1647 +	0.3496 +	0.1499 +	0.6639 +	0.6843 +	0.5320 +	0.6685 +
BHVmu	0.3511 +	0.2285 +	0.3885 +	0.3517 +	0.9930 −	1.0471 −	1.0420 −	1.0128 −

Positive was shown with + and negative was shown with −.

## Data Availability

Data supporting the reported results are available in this article.

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
