# Peer review of "Application of CRISPR/Cas9 for Rapid Genome Editing of Pseudorabies Virus and Bovine Herpesvirus-1"

_viruses, 2024, doi:10.3390/v16020311_

Round 1

Reviewer 1 Report

Comments and Suggestions for Authors

The manuscript “Application of CRISPR/Cas9 for rapid genome editing of Pseudorabies Virus and Bovine Herpesvirus-1” by Yu et al. describes the use of CRISPR/Cas9 technology for gene deletion and insertion of eGFP to show protein expression. Although this technology has been used to edit several herpesviruses in poultry and mammals and is used for recombinant vaccines. 

I have the following minor comments related to the manuscripts:

  1. There are several papers that have used CRISPR/Cas technologies to edit BHV-1. e.g., PMID: 35755997; 35873151. Both of these publications are from China.
  2. Be consistent in using BHV-1 in several places where BHV was used (line 152, BHV mu instead of BHV-1 mu); similarly, lines 163, 224, etc. Figure 4D)
  3. Any back passage was done to see whether the incomplete gE deletion reverted back to the wild type.

Author Response

Thank you very much for taking the time to review this manuscript. Please find the detailed responses below and the corresponding revisions highlighted in the re-submitted files.

Comments 1: There are several papers that have used CRISPR/Cas technologies to edit BHV-1. e.g., PMID: 35755997; 35873151. Both of these publications are from China.

Response 1: Thank you for pointing this out. We have included these publications in the Introduction section (line 68 of the newest version).

Comments 2: Be consistent in using BHV-1 in several places where BHV was used (line 152, BHV mu instead of BHV-1 mu); similarly, lines 163, 224, etc. Figure 4D)

Response 2: Thank you for pointing this out. It is correct to use BHV-1 consistently where it refers to the virus type. Accordingly, we have modified “BHV” to “BHV-1” in line 16, 245, 246, 248, 287 and Figure 4D. For the case of BHVmu, it is just a shortened name we have given to our mutant. So we kept it in our manuscript in line 155, 163, etc..

Comments 3: Any back passage was done to see whether the incomplete gE deletion reverted back to the wild type.

Response 3: In our experiment, we sequenced the genomic DNA of the 14th passage of BHV-1 mutant, and no reversion was found. On the one hand, we have removed most of the gE gene, leaving only a small number of C-terminal bases, and on the other hand, the DNA virus is relatively stable. These all reduce the probability of the mutant reverting to the wild type form.

Reviewer 2 Report

Comments and Suggestions for Authors

This study developed a rapid and straightforward approach to manipulating herpesviruses (PRV and BHV-1) genome using CRISPR/Cas9. It is well written and would be beneficial for the vaccine development. However, there are some minor concerns.

1.     Line 231-234, as described PRV Bartha K61, a vaccine strain, is lethal to mice, PRVmu can also cause mice deaths. Are there any other methods to evaluate the effect of attenuation?

2.     Line 77 and 85, GeneBank should be GenBank.

3.     Line 124, TCID50 5×107 should be TCID50=5×107

Author Response

Thank you very much for taking the time to review this manuscript. Please find the detailed responses below and the corresponding revisions highlighted in the re-submitted files.

Comments 1: This study developed a rapid and straightforward approach to manipulating herpesviruses (PRV and BHV-1) genome using CRISPR/Cas9. It is well written and would be beneficial for the vaccine development. However, there are some minor concerns.

Line 231-234, as described PRV Bartha K61, a vaccine strain, is lethal to mice, PRVmu can also cause mice deaths. Are there any other methods to evaluate the effect of attenuation?

Response 1: Since PRV can infect several mammals, including rodents, bovine, and porcine[1], many researchers choose mice as the model to evaluate the pathogenicity of PRV attenuated mutants[2][3]. In this species several tool are available to detect interferons and other cytokines during PRV infection to evaluate the effect of attenuation. Due to the main focus of our manuscript being the technology to modify herpesvirus, we did not characterize further the effect of the attenuation through complementary approaches. This will certainly be the focus of the next studies where we will evaluate the suitability of our new mutant starins as potential vaccine vectors.

References:

  1. Mettenleiter, T.C. Aujeszky’s Disease (Pseudorabies) Virus: The Virus and Molecular Pathogenesis - State of the Art, June 1999. Vet. Res. 2000, 31, 99–115, doi:10.1051/vetres:2000110.
  2. Tang, Y.-D.; Liu, J.-T.; Wang, T.-Y.; Sun, M.-X.; Tian, Z.-J.; Cai, X.-H. Comparison of Pathogenicity-Related Genes in the Current Pseudorabies Virus Outbreak in China. Sci. Rep. 2017, 7, 7783, doi:10.1038/s41598-017-08269-3.
  3. Ren, J.; Tan, S.; Chen, X.; Yao, J.; Niu, Z.; Wang, Y.; Ma, L.; Gao, X.; Niu, S.; Liang, L.; et al. Genomic Characterization and GE/GI-Deleted Strain Construction of Novel PRV Variants Isolated in Central China. Viruses 2023, 15, doi:10.3390/v15061237.

Comments 2: Line 77 and 85, GeneBank should be GenBank

Response 2: Thank you for pointing this out. We have corrected these at line 94 and 112 of the newest version.

Comments 3: Line 124, TCID50 5×107 should be TCID50=5×107.

Response 3: We have corrected it in line 142.

Reviewer 3 Report

Comments and Suggestions for Authors

This appears to be a relatively sound piece of work and the authors have shown that the techniques described produced the desired outcomes.

Comments on the Quality of English Language

The authors appear to fail to realise that the word “virus” is a noun under word “viral” is the appropriate adjective.

These two words appear to be randomly used throughout much of the text.

Line 62

Should read Alphaherpesvirus gene editing that can be used to rapidly construct

Line 91

Should read sequences were chosen from upstream and downstream

Line 114

Generation of viral mutants

Line 156

Supernatant was collected for viral titration

Note that in the same line the authors used the adjective correctly.

Line 169

The acronym ELISA stands for enzyme-linked immunosorbent assays. The term ELISA assay is tautology.

Line 197

to viral rescue

Line 210

Viral replication

Line 248

This is probably mouse body weight not mice bodyweight.

Line 237

Analysed by two-way ANOVA and shown with

On this page the words virus and viral appear to be used randomly

Line 275 viral biology

280 viral strains

287 viral biology

Line 314

The term in vivo should be italicised

Line 318 and 319

Gene of interest using the same method

Author Response

Thank you very much for taking the time to review this manuscript. Please find the detailed responses below and the corresponding revisions highlighted in the re-submitted files.

Response to Comments on the Quality of English Language

Point 1: The authors appear to fail to realise that the word “virus” is a noun under word “viral” is the appropriate adjective. These two words appear to be randomly used throughout much of the text.

Response 1: Thanks for pointing this out. For the case of “virus genome”, we have modified “virus” to “viral” in line 101 and 244 (of the newest version). For “virus mutants” in line 132, “virus titration” in line 174, “Virus infectivity” in line 176, “virus rescue” in line 216, “recombinant virus replication” in line 229, “virus mutant” in line 247, “virus attenuation” in line 256, “virus growth curve” in line 260, “virus biology” in line 294 and 306, and “virus strains” in line 299, we still kept them as the compound nouns which are common in our field.

Point 2: Line 62. Should read Alphaherpesvirus gene editing that can be used to rapidly construct

Response 2: We have added “can” in our manuscript in line 79.

Point 3: Line 91. Should read sequences were chosen from upstream and downstream

Response 3: We have deleted “the” in line 119.

Point 4: Line 114. Generation of viral mutants

Response 4: Thanks for the advice. We would like to keep “virus mutants” as a compound noun, which is also marked in Point 1. This puts the emphasis on the specific viruses we were targeting.

Point 5: Line 156. Supernatant was collected for viral titration. Note that in the same line the authors used the adjective correctly.

Response 5: Similar reason as point 1 and point 4. And for the case of “viral titers”, we wanted to emphasize the “titers”, so we use “viral” as an adjective.

Point 6: Line 169. The acronym ELISA stands for enzyme-linked immunosorbent assays. The term ELISA assay is tautology.

Response 6: We have added the complete term in line 188.

Point 7: Line 197. to viral rescue. Line 210. Viral replication.

Response 7: Already mentioned at point 1.

Point 8: Line 248. This is probably mouse body weight not mice bodyweight.

Response 8: We have modified it in line 279.

Point 9: Line 237. Analysed by two-way ANOVA and shown with

Response 9: Thank you, we have amended this error.

Point 10: On this page the words virus and viral appear to be used randomly.

Line 275 viral biology

280 viral strains

287 viral biology

Response 10: Already mentioned in point 1.

Point 11: Line 314. The term in vivo should be italicised

Response 11: We have modified it in line 333.

Point 12: Line 318 and 319. Gene of interest using the same method

Response 12: We have deleted “by” in lines 337 and 338.